# Treadmill Exercise Reverses the Adverse Effects of Intermittent Fasting on Behavior and Cortical Spreading Depression in Young Rats

**DOI:** 10.3390/brainsci13121726

**Published:** 2023-12-17

**Authors:** Amanda Ferraz Braz, Maria Luísa Figueira de Oliveira, Dominique Hellen Silva da Costa, Francisco Leonardo Torres-Leal, Rubem Carlos Araújo Guedes

**Affiliations:** 1Department of Nutrition, Federal University of Pernambuco, Recife 50670-901, PE, Brazil; 2Department of Physiology and Pharmacology, Federal University of Pernambuco, Recife 50670-901, PE, Brazil; 3Metabolic Diseases, Exercise and Nutrition Research Group (DOMEN), Department of Biophysics and Physiology, Federal University of Piauí, Teresina 64049-550, PI, Brazil

**Keywords:** periodic fasting, physical activity, anxiety-like behavior, memory, electrophysiological phenomenon

## Abstract

Intermittent fasting (IF) and physical exercise (PE) have beneficial psychological and physiological effects, improving memory and anxiety-like behavior. However, the impact of this combination on brain electrophysiological patterns is unknown. We aimed to evaluate the behavior and parameters of a brain excitability-related phenomenon named cortical spreading depression (CSD) in young rats (31–87 days of life) submitted to IF and treadmill PE for eight weeks. Sixty-four male and female Wistar rats aged 24 days were randomized into control, IF, PE, and IF+PE groups. Behavioral tests (open field (OF), object recognition, and elevated plus maze (EPM)) were performed, and the CSD propagation features were recorded. IF caused behavioral responses indicative of anxiety (lower number of entries and time spent in the OF center and EPM open arms). IF also reduced the discrimination index for object recognition memory tests and increased the propagation velocity of CSD. PE rats displayed more entries into the OF center and lowered CSD propagation speed. Data suggest that IF worsens anxiety-like behavior and memory and accelerates CSD in young rats. In contrast, PE reverted the unfavorable effects of IF. The brain effects of IF and PE at younger ages are recommended for study.

## 1. Introduction

Intermittent fasting (IF) is a clinical and experimental feeding strategy in which periods of restricted or absent caloric intake are interspersed with periods of unlimited consumption [1]. This strategy is based on an old voluntary or obligatory practice used worldwide for religious and health-improving purposes [2]. Regular IF and physical exercise (PE) induce positive psychological and physiological effects, contributing to mental and physical health [3].

IF and PE’s “metabolic switch” has been implicated in regulating the brain’s functions related to neuroplasticity, learning, and memory capacity [4]. Treadmill exercise increased open arm entries in elevated plus maze in mice, suggesting the amelioration of anxiety-like behavior [5]. PE in rodents acts on the release of serotonin and improves synaptic transmission, promoting anxiolytic effects [6]. Furthermore, PE, through mechanisms such as angiogenesis, autophagy, and the reduction of inflammatory markers, also contributes to the delay of brain aging and preservation of memory [7].

Data on the behavioral effects of IF are limited and sometimes controversial; a study in mice subjected to IF demonstrated enhanced memory, consolidating long-term memories and improving dentate gyrus neurogenesis [8]. Different IF protocols attenuated anxiety-like behavior in healthy and colitis mice and alleviated neuroinflammation and oxidative stress [9]. However, in a human meta-analysis, it was demonstrated that IF did not modify the scores related to anxiety or mood [10].

There is a lack of studies relating IF behavioral changes to electrophysiological patterns assessed by the excitability-related phenomenon known as cortical spreading depression (CSD). CSD consists of a depolarizing “wave-like” reduction (depression) in spontaneous and evoked neuronal activity elicited by an electrical, mechanical, or chemical stimulus at one point of the cerebral cortex. This reversible response spreads slowly concentrically to remote cortical regions while the eliciting point recovers [11]. Considering that the brain’s electrical activity controls its primary functions, studying CSD represents an essential tool for understanding brain functioning in health and disease [12].

A series of nutritional and non-nutritional variables of clinical interest have already been identified as facilitating or hindering the spread of CSD and influencing behavior [12]. However, more information is needed regarding the implications of combining IF and PE on CSD. Thus, our primary goal was to evaluate the impact of IF and PE on the behavioral parameters of anxiety and short-term memory retention and CSD electrophysiological parameters.

## 2. Material and Methods

### 2.1. Ethical and Animal Aspects

This study was approved by the Ethical Committee for using animals in scientific research of the Federal University of Pernambuco (protocol no. 006/2021, 2 July 2021), whose norms comply with the norms established by the National Institutes of Health Guide for Care and Use of Laboratory Animals (Bethesda, MD, USA). A total of 64 Wistar rats of both sexes (30 males and 34 females) aged 24 days were obtained from our department’s vivarium. Animals were separated by sex, housed in polypropylene cages (51 cm × 35.5 cm × 18.5 cm) with four rats per cage, and maintained in an environment with controlled conditions of temperature (23 ± 1 °C) and lighting (light-dark cycle of 12 h, with the lights on at 6:00 a.m.). All efforts were made to minimize animal suffering and to use the minimum number of animals to obtain valid results.

### 2.2. Experimental Protocol

Animals were randomly distributed into four groups: (a) the control, which was sedentary and with free access to a laboratory chow diet (*n* = 14, of which 6 were males) and were not subjected to intermittent fasting or to exercise; (b) intermittent fasting (IF) (*n* = 20; 10 males); (c) physical exercise (PE) (*n* = 15; 7 males); and (d) intermittent fasting + physical exercise (*n* = 15; 7 males). We used a physical method of randomization: pieces of paper with the group’s names written on them were placed in a receptacle, mixed, and withdrawn repeatedly, so each rat was allocated to one of the four groups. First, the animals were familiarized with the study protocols for one week. After that, the main experiment of IF and PE on a treadmill lasted eight weeks. Body weight and food intake were monitored weekly and recorded on postnatal days (PND) 33, 57, and 80. At last, the animals were subjected to behavioral tests to evaluate anxiety and memory, followed by the recording of CSD and, after that, euthanasia. Figure 1 presents the time diagram of the various experimental procedures and their respective ages.

### 2.3. Intermittent Fasting

All rats were fed a standard laboratory chow diet (Nuvilab^®^, with 25% protein; Quimtia, Colombo, Paraná State, Brazil) and had free access to water. During six days (PND 25–30), animals were familiarized with the feeding/fasting regimes inserted. In the IF group, fasting was started at PND 31 during 24 h for three non-consecutive d/wk [1]; food was removed from the cage at 08:00 a.m. to start the fasting day and then put back in the cage at the same time the following day to begin the feeding day. The fast was interrupted at the end of eight weeks, and the standard lab chow diet was then freely available. Body weight and food intake were monitored weekly at the same time in the morning. Food intake was determined from the average consumption found for each cage.

### 2.4. Exercise Program

Rats were distributed into sedentary and exercised groups; those in the exercise group were familiarized with the treadmill for six days (from PND 25 to 30). Physical exercise was used for rodents (running on a motorized treadmill; INSIGHT, model EP-131). Each familiarization session included placing the rats on the treadmill switched off for 10 min and then turning on the treadmill at 8 m/min for 5 min [13]. Moderate-intensity exercise was performed in sessions of 40 min a day, three times a week, for eight weeks (from PND31 to 87), according to the following protocol [14]: 5 min warming up at a speed of 12 ± 2 m/min, 30 min at a main speed of 20 ± 2 m/min, and 5 min cooling down at a speed of 12 ± 2 m/min. No aversive stimulus was used, and the rats that refused to run were stimulated with gentle touches; if persistent in refusing, they were discarded from the study. Animals from the sedentary groups were placed on the treadmill switched off for the same period as the trained animals in each session [15].

### 2.5. Identification of Estrous Cycle

To standardize the hormonal physiological environment of the females, before the behavioral tests and the CSD recording, the phase of the estrous cycle was identified. To this, the smear technique was performed by collecting vaginal mucosal cells as previously described [16]. Vaginal secretions were collected with a flexible rod with a cotton tip soaked with 0.9% NaCl. The smear was transferred to a clean glass slide and visualized in a light microscope for cytology. Three cell types were used for determination: epithelial cells, cornified cells, and leucocytes. Proestrus was mainly characterized by nucleated epithelial cells and was a determinant for conducting behavioral tests and CSD recording procedures [16].

### 2.6. Behavioral Tests

The test room had sound attenuation and low light intensity. Before each test, the animals remained in the room for 20 min to adapt to the environment. All rat movements were recorded for 5 min of the test by a digital camera located vertically above the test device. The device was cleaned with a 70% ethanol–water solution between the sessions to minimize distinct olfactory signals. The video-recorded activity was stored on a computer and analyzed using the ANYmaze^®^ software (4.99 m version) [11].

### 2.7. Open Field Test

Rats were placed in the center of a circular arena (89 cm diameter) surrounded by a wall (52 cm height) made of wood. The floor was painted with lines to distinguish four quadrants, and the center was defined by a circle (62 cm diameter). The evaluated parameters were distance traveled, immobility time, central zone entries, and central zone time [17].

### 2.8. Object Recognition Tests

Novel object recognition tests were evaluated as described previously [15] and were tested in the OF arena. In the first session (training session), rats explored two identical objects for 5 min. These objects were placed in the arena at equal distances from the wall in an asymmetric position regarding the center. After a 40 min inter-session interval, the rats were returned to the arena (test session). They had to recognize that one of the objects was moved to a novel spatial position (day 1) or that another object had replaced one with a different shape (day 2). Based on the exploration times on novel (N) and familiar (F) objects/positions, we calculated the discrimination index (DI) using the formula DI = (TN − TF)/(TN + TF), where TN and TF are the time spent with the novel object/position and the familiar object/position, respectively [15].

### 2.9. Elevated plus Maze Test

The EPM apparatus was cross-shaped and made of varnished wood, elevated 55 cm above the floor, with two open arms and two closed arms, each measuring 49 cm long × 10 cm wide. The closed arms, with side walls measuring 50 cm high, were arranged perpendicular to the open ones. A central 10 × 10 cm square platform joined the arms of the apparatus. The test began with each rat placed individually in the central platform, facing one of the open arms, and freely exploring the maze for 5 min [17]. The parameters considered were the distance traveled, immobility time, open arm entries, and open arm time [17].

### 2.10. CSD Recording

On PND 94-100, each animal was weighed and anesthetized with an intraperitoneal injection of a mixture containing 1 g/kg urethane and 40 mg/kg chloralose. The head of the rat was secured in a stereotaxic apparatus (David Kopf Instruments, Tujunga, CA, USA), and three trephine holes (2–4 mm diameter) were drilled on the right side of the skull aligned in the frontal-to-occipital direction and parallel to the midline. One hole was positioned on the frontal bone and was used to apply the stimulus to elicit CSD. The other two holes were drilled in the parietal bone and were used to record the propagating waves. At 20 min intervals, CSD was elicited by a 1 min application of a cotton ball (1–2 mm diameter) soaked with a 2% KCl solution to the anterior hole. The ECoG depression and the direct current (DC) slow potential change accompanying CSD were recorded for 4 h on the cortical surface through a digital recording system (Biopac MP 150, Goleta, CA, USA) [18]. Rectal temperature was continuously recorded and maintained at 37 °C ± 1° C. The CSD propagation was monitored in the two exposed cortical portions on the parietal surface of the cerebral cortex using two Ag–AgCl electrodes against a common reference electrode of the same type placed on the nasal bones. The electrodes consisted of chlorided silver wires inserted into plastic pipettes (5 cm long, 5 mm diameter at the opening top, and 0.5 mm inner diameter at the tip) filled with Ringer’s solution solidified with the addition of 0.5% agar. We calculated the CSD velocity of propagation based on the time spent by a wave to pass the distance between the two cortical recording electrodes gently placed on the dura mater. We also calculated the amplitude and duration of the DC slow potential change of the CSD waves.

### 2.11. Statistical Analysis

Results are expressed as mean ± standard deviation (SD). The statistical software used was Sigmastat version 3.10. A *p*-value < 0,05 was considered statistically significant. A two-way ANOVA test (considering intermittent fasting and physical exercise as factors) was used, followed by the Holm–Sidak post hoc test, to identify significant contrasts.

## 3. Results

### 3.1. Body Weight and Food Intake

The animals’ body weight and food intake were recorded and analyzed at three time points during the experiment: PND 33, 57, and 80 (the initial, intermediate, and final periods, respectively). Our findings confirmed data from the literature regarding a higher body weight (Table 1) and greater food ingestion in male rats than females (Table 2).

As all groups contained nearly an equal number of males and females and considering that no difference between sexes was observed in the other parameters (behavior and CSD), we analyzed those parameters by including males and females in a single group. ANOVA revealed a main effect of IF on body weight at PND 33 (F[3, 60] = 10.143; *p* = 0.002), PND 57 (F[3, 64] = 8.808; *p* = 0.004) and PND 80 (F[3, 60]= 4.478; *p* = 0.038). The Holm–Sidak post hoc test indicated that the IF animals displayed lower weights than the control animals (*p* < 0.05).

PE mainly affected body weight at PND 33 (F[3, 60 = 4.781; *p* = 0.033), and the Holm–Sidak post hoc test showed that the exercised group had higher body weights than the sedentary group (*p* < 0.05).

Regarding the food intake, ANOVA demonstrated that IF reduced the amount of food ingested at PND 33 (F[3, 60] = 37.011; *p* < 0.001), at PND 57 (F[3, 60] = 18.934; *p* < 0.001), and at PND 80 (F[3, 60] = 14.900; *p* < 0.001). In contrast, PE increased body weight at PND 33 (F[3, 60] = 8.216; *p* = 0.006).

### 3.2. Anxiety-like Behavior

The effects of IF and PE on anxiety-like behavior in the open field (OF) test are in Figure 2. The parameters of traveled distance (Figure 2A) and immobility time (Figure 2B) did not suffer interference from the studied variables. The IF rats displayed fewer entries into the center (F[3, 60] = 17.328; *p* < 0.001) and spent a shorter time in the center of the open field (F[3, 60] = 5.646; *p* = 0.021)). On the other hand, PE increased the number of entries into the center (F[3, 60] = 5.646; *p* = 0.021). There was no statistical difference in time spent in the center between the PE and other groups (Figure 2C,D).

Anxiety-like behavior was also observed in the elevated plus maze (EPM) test (Figure 3). Similar to OF, groups did not differ for traveled distance and immobility time (Figure 3A,B). ANOVA revealed an interaction between IF and PE regarding the number of entries into the open arms (F[3, 60] = 20.098; *p* < 0.001) and the time spent in the open arms (F[3, 60] = 9.101; *p* = 0.004). Intermittent fasting reduced the number of entries and time spent in the open arms compared to the C group and IF+PE group (*p* < 0.05). PE was associated with increased entries into the open arms and a long duration of time spent in the open arms, which was more intense in the IF+PE group. However, although significantly higher than the IF group, it was statistically equal to the C group (Figure 3C,D).

### 3.3. Object Recognition Memory

The data of the shape and spatial position recognition tests are shown in Figure 4. ANOVA revealed that IF reduced short-term memory retention, as evaluated by the reduced discrimination index for novel shape (F[3, 60] = 4.191; *p* = 0.045) and spatial position (F[3, 60] = 9.764; *p* = 0.003) compared to the C and IF+PE groups (Figure 4A,B). The PE group did not differ from the others.

### 3.4. CSD Features

The topical application of a cotton ball soaked with 2% KCl on the frontal cortex promoted a single CSD wave that propagated and was recorded by two electrodes located on the parietal bone. A third electrode positioned on the nasal bone was a common reference for the other two. The recording of the slow voltage change and the reduction in the electrocorticogram confirmed CSD (Figure 5A).

IF significantly accelerated the CSD propagation compared to the C group (F[3, 60] = 27.631; *p* < 0.001), while PE slowed down the spread of CSD (F[3, 60] = 188.559; *p* < 0.001, Figure 5B). Similarly, the amplitude of CSD was greater in IF compared to the C group (F[3, 60] = 23.589; *p* < 0.001). However, the PE groups exhibited smaller CSD amplitudes (F[3, 60] = 209.387; *p* < 0.001, Figure 5C). The CSD’s duration was significantly reduced by IF (F[3, 60] = 25.639; *p* < 0.001) but increased by PE (F[3, 60] = 145.728; *p* < 0.001, Figure 5D).

## 4. Discussion

The present study provides evidence of the role of IF combined with PE on anxiety and memory behavior and the brain excitability-related CSD phenomenon. Our main findings suggest that when started early in life, IF worsens anxiety-like behavior and memory. Furthermore, IF makes the brain more likely to propagate CSD by accelerating it. On the other hand, PE reverses the harmful effects of IF by improving anxiety-like behavior and memory and slowing down CSD propagation. To our knowledge, this is the first study analyzing the interaction of IF and PE on anxiety behavior, memory, and CSD in developing rats.

We have currently demonstrated that animals subjected to IF had reduced food intake and body weight. Similar results were also previously observed in adult rats [19], suggesting that IF could be an exciting strategy for helping treat adult obesity. PE did not promote changes in the body weight of animals compared to the control group, as previously demonstrated [20], and the increase in food consumption probably compensated for the higher energy expenditure resulting from the exercise.

The data obtained from anxiety-like behavior tests demonstrated that the animals from the IF group entered less and spent less time in the center of the OF test(Figure 2) and the open arms of EPM (Figure 3), which suggests an anxiogenic effect of IF at this age. Moreover, the reduction in the discrimination index in short-term memory tests (Figure 4), that is, less exploration of new spatial positions and objects, suggests a worsening of memory caused by IF.

The effects of IF on behavior observed here contrast with some evidence in the literature. Eight-week-old female mice subjected to IF for three months exhibited improved long-term memory retention [8]. On the other hand, every-other-day fasting for six weeks did not affect memory formation in male rats [21]. Sixteen hours of food deprivation daily for three months reduced anxiety-like behavior in diabetic male rats aged 12–14 weeks [22]. Twenty-four-hour IF every other day for 28 days improved the spatial memory of three-month-old diabetic mice [23]. Mice (7–8 weeks old) with colitis had attenuated anxiety symptoms after 16 h of IF for 36 days [9]. These contrasting results can be attributed to methodological factors, such as different IF regimens, animal species and ages, and associated pathologies. IF comprises an umbrella term surrounding other dietary restriction protocols, making it difficult to standardize the methodology and provide heterogeneous results. In addition, most studies use experimental models of associated pathology, contrasting with the clinically healthy animals used here.

Although mice and rats are evolutionarily similar, there are significant differences between the species regarding morpho-functional brain features, including, for example, dendritic length, dendritic spine size, and resting membrane potential, which makes direct comparisons difficult [24]. Even though several studies with rodents show beneficial effects of IF on memory, anxiety-like behavior, and neurological diseases [25], the evidence remains discreet and conflicting in humans. Ramadan, for example, was responsible for improving or not interfering with anxiety-like behavior, well-being, or mood and promoting relapses in episodes of bipolar disorder and worsening symptoms of schizophrenia [10,26,27]

Another factor that may contribute to the diversity of results is the age of the animals. Unlike most studies in the literature, which use adult or older animals, our analysis focused on the effects of IF in the development period. This phase is considered critical for neurodevelopment since the nervous system develops over a long period, from embryonic to puberty, in rats and humans [28]. Environmental factors, including dietary aspects, can interfere with brain development during early life, bringing lasting and often irreversible effects to cognitive development and mental health. Therefore, the effects of IF and PE on a developing organism might be considered differently from those effects on the developed brain [29]. While IF and PE appear beneficial in adulthood, the same is not in developing organisms.

Furthermore, maternal nutrition during pregnancy profoundly affects the growth and development of offspring throughout life [30]. However, the role of IF during pregnancy on fetal development and health is not yet well defined. Even though pregnant Muslim women are exempt from fasting during Ramadan, many of them still choose to participate [31]. For this reason, some animal studies replicate this model of Ramadan fasting during pregnancy. The reported effects are related to an altered profile of placental metabolites, reduced transport of placental amino acids, and fetal growth restriction [32,33]. Impairment of placental function and fetal growth can interfere with development with potentially long-lasting effects on offspring physiology later in life and a greater predisposition to diseases [33].

Our findings also point out the action of IF in modulating brain excitability, increasing CSD’s propagation velocity. As far as we know, this is the first report of the accelerating effect of IF on the CSD phenomenon in young rats. The closest available literature is the effect of acute 24 h fasting on the increase in the excitability of neural activity, specifically in the olfactory bulb of eight-week-old mice [34]. Conversely, ketogenic diets and IF are associated with reduced neuronal excitability in animals [35], which differs from the present data. Since the results are controversial and the exact mechanisms still need to be fully determined, further studies are required. Nutritional deficiency early in life facilitates CSD propagation in the rat cortex (12). In the present study, the lower body weight and food intake observed in the IF group (Table 1 and Table 2) suggest a certain degree of nutritional deficiency, which is in line with CSD facilitation in these animals (Figure 5).

On the other hand, there is a growing body of scientific evidence that PE promotes beneficial effects on physiological and mental health in humans and animal models [36,37]. In line with the literature [5,38], our findings indicated that treadmill PE associated with IF improved memory and anxiety-like behavior in rats compared to IF (see Figure 3D). The CSD phenomenon also shows a clear effect of exercise (see Figure 5B). The beneficial effect of our moderate exercise paradigm on behavior was less than one could expect from the literature data. A more intense or lasting exercise paradigm would probably produce more conspicuous effects on behavior.

Potential mechanisms involved in the benefits of PE on brain activity include neurogenesis and angiogenesis in the hippocampus and the activation and modulation of neurotrophins and growth factors [39]. PE enhances the generation of new neurons and stimulates the formation of new vessels, which contributes to memory and spatial learning [40]. Thus, PE is receiving more interest as a non-drug method of maintaining brain health and treating neurodegenerative and psychiatric conditions.

Those mechanisms mentioned above also help to understand the role of PE in the brain’s electrophysiological activity based on the analysis of CSD. By stimulating neurogenesis, angiogenesis, and synaptogenesis, PE improves local blood flow and increases the distance to be covered by cell–cell communication, possibly influencing brain excitability and slowing down CSD [41]. These data agree with a previous study showing that treadmill PE can antagonize the spread of CSD in rats [20].

Summarily, IF in young rats worsens anxiety-like behavior and memory and accelerates CSD, while PE improves the behavioral aspects of anxiety and memory and slows CSD. Furthermore, PE, when associated with IF, could reverse its adverse effects on the brain, similar to those previously demonstrated [42]. Therefore, these findings indicate that combining such factors is the most stated choice since PE interacts with dietary elements, modulating energy metabolism, and synaptic plasticity, which promotes positive effects on brain functioning.

## 5. Conclusions

Our findings suggest that IF worsens anxiety-like behavior and memory and accelerates CSD when initiated in developing rats. We reinforce the protective effect of PE on the brain, improving anxiety-like behavior and memory and slowing down CSD. These results indicate that the association between IF and PE may be necessary for brain health since PE can reverse IF’s harmful effects. However, the underlying mechanisms still need to be fully elucidated, especially at younger ages, being necessary for further investigations.

## Figures and Tables

**Figure 1 brainsci-13-01726-f001:**
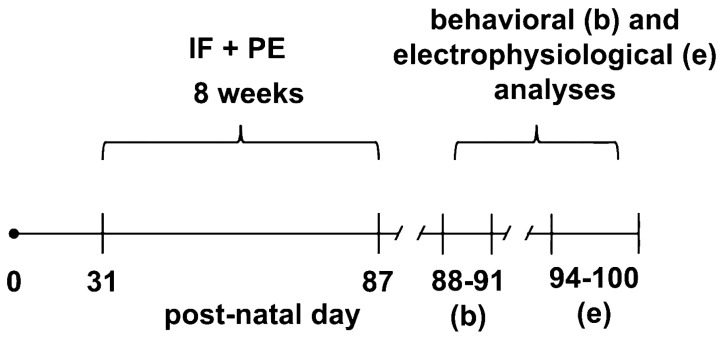
Time diagram showing the various experimental procedures and the respective ages when they occurred. IF = intermittent fasting; PE = physical exercise. We classified the animals as young because the rats were subjected to IF and PE at 31 to 87 PND.

**Figure 2 brainsci-13-01726-f002:**
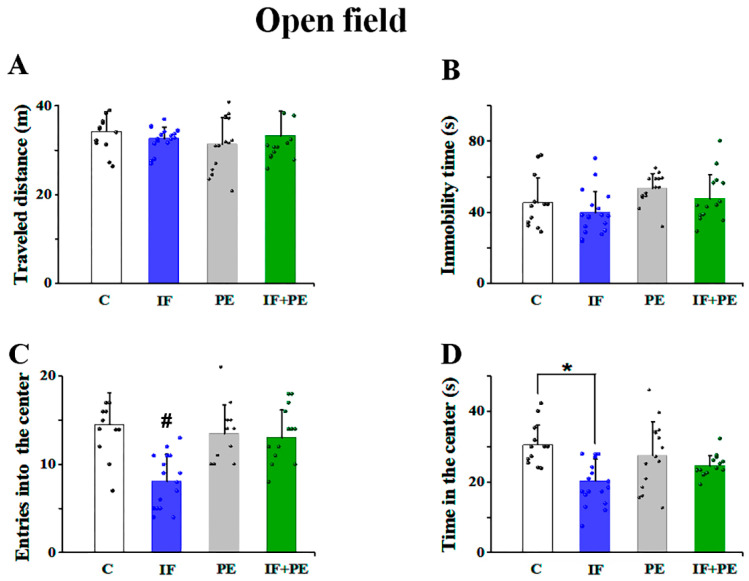
Anxiety-like behavior in the open field test. (**A**) Distance traveled in meters. (**B**) Immobility time in seconds. (**C**) Number of entries into the center. (**D**) Time spent in the center. Values are mean ± SD of 14–20 animals per group. # Significantly different from the other three groups. * *p* < 0.001, two-way ANOVA followed by the Holm–Sidak post hoc test.

**Figure 3 brainsci-13-01726-f003:**
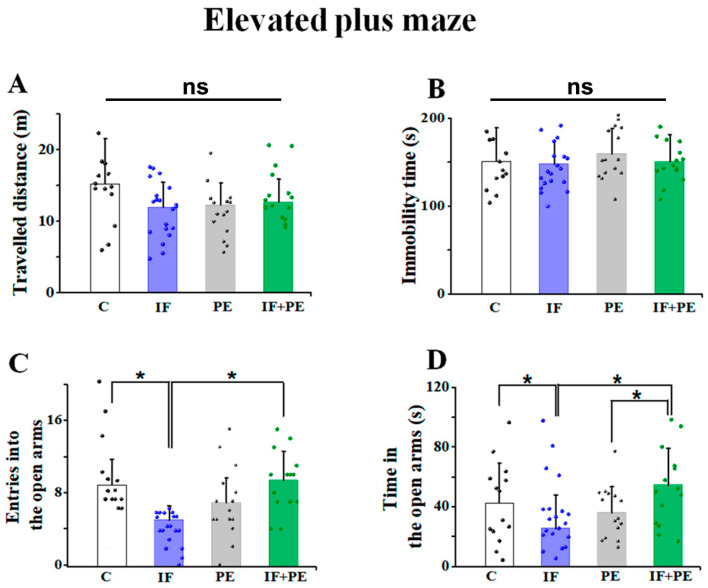
Anxiety-like behavior in the elevated plus maze (EPM) test. (**A**) Distance traveled in meters. (**B**) Immobility time in seconds. (**C**) Number of entries into the open arms. (**D**) Time spent in the open arms. Values are mean ± SD of 14–20 animals per group. Non-significant differences are indicated by ns. * *p* < 0.05, two-way ANOVA, followed by the Holm–Sidak post hoc test.

**Figure 4 brainsci-13-01726-f004:**
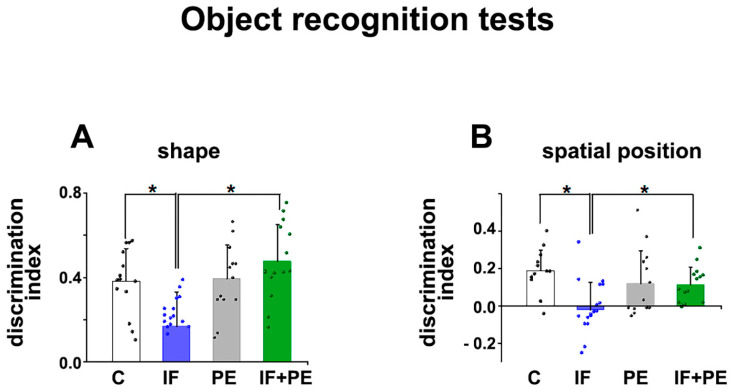
Memory evaluation by object recognition tests. (**A**) Discrimination index for the shape recognition test. (**B**) Discrimination index for the spatial position recognition test. Values are mean ± SD of 14–20 animals per group. * *p* < 0.05, two-way ANOVA followed by the Holm–Sidak post hoc test.

**Figure 5 brainsci-13-01726-f005:**
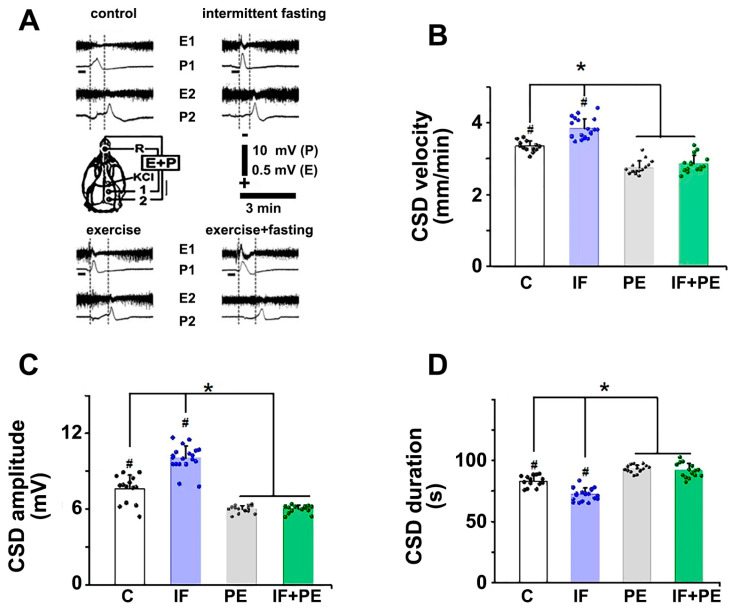
Qualitative (**A**) and quantitative representations (**B**–**D**) of cortical spreading depression (CSD) propagation. (A) Electrocorticogram (E) and slow potential change (P) at two points on the surface of the right hemisphere. The skull diagram shows the positions 1 and 2 of the recording electrodes, the common reference electrode (R) position on the nasal bone, and the CSD-eliciting stimulus (KCl) application point. The dashed vertical lines indicate the latency of the CSD to cross the distance between the electrodes, and the horizontal bar represents the 3 min time scale. (**B**) CSD velocity in mm/min. (**C**) CSD amplitude in millivolts. (**D**) CSD duration in seconds. Values are mean ± SD of 14–20 animals per group. # Significantly different from the other three groups. * *p* < 0.001, two-way ANOVA followed by the Holm–Sidak post hoc test.

**Table 1 brainsci-13-01726-t001:** Body weights (g; mean ± standard deviation) of 30 male (M) and 34 female rats (F) randomly allocated in the following four experimental groups: control (*n* = 14, from which six were males), intermittent fasting (*n* = 20; 10 males), physical exercise (*n* = 15; 7 males) and intermittent fasting+physical exercise (*n* = 15; 7 males). Weights were evaluated on PND 38, 57, and 80. * Significantly different from the corresponding female values (*p* < 0.05; ANOVA followed by the Holm–Sidak post hoc test).

Group	38 d		57 d		80 d	
	M *	F	M *	F	M *	F
Control (*n* = 14; 6 M)	155.0 ± 22.3	118.5 ± 16.5	258.3 ± 19.5	168.9 ± 7.4	298.5 ± 17.4	191.4 ± 12.1
IF (*n* = 20; 10 M)	128.8 ± 21.2	104.3 ± 11.4	194.5 ± 21.6	152.3 ± 12.4	231.1 ± 23.7	176.5 ± 13.0
PE (*n* = 15; 7 M)	181.3 ± 19.1	130.0 ± 12.4	277.7 ± 27.0	179.5 ± 16.4	327.3 ± 52.1	196.8 ± 15.7
IF+PE (*n* = 15; 7 M)	136.1 ± 17.2	105.4 ± 17.4	215.6 ± 21.0	150.5 ± 14.4	260.7 ± 25.5	178.0 ± 11.3

**Table 2 brainsci-13-01726-t002:** Food intake (g; mean ± standard deviation) of 30 male (M) and 34 female rats (F), as described in Table 1 above. * Significantly different from the corresponding female values (*p* < 0.05; ANOVA followed by the Holm–Sidak post hoc test).

Group	38 d		57 d		80 d	
	M *	F	M *	F	M *	F
Control (*n* = 14; 6 M)	17.8 ± 0.8	13.9 ± 1.2	20.2 ± 0.1	14.5 ± 0.3	18.2 ± 2.3	13.6 ± 0.9
IF (*n* = 20; 10 M)	13.0 ± 2.7	9.8 ± 1.1	15.9 ± 4.6	13.2 ± 2.5	14.7 ± 0.7	12.4 ± 1.3
PE (*n* = 15; 7 M)	21.1 ± 0.5	17.8 ± 5.9	21.5 ± 1.9	15.5 ± 1.4	20.3 ± 1.8	14.2 ± 1.6
IF+PE (*n* = 15; 7 M)	12.8 ± 1.1	10.6 ± 1.3	15.5 ± 1.4	11.0 ± 1.5	16.1 ± 1.8	11.8 ± 0.7

## Data Availability

The data supporting this study’s findings are available from the corresponding author upon reasonable request. Data are not publicly available because dissemination has yet to be explicitly foreseen by the local ethics committee.

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
