# Peer review of "Treadmill Exercise Reverses the Adverse Effects of Intermittent Fasting on Behavior and Cortical Spreading Depression in Young Rats"

_brainsci, 2023, doi:10.3390/brainsci13121726_

Round 1

Reviewer 1 Report

Comments and Suggestions for Authors

1. Provide a more comprehensive description of the intermittent fasting (IF) and physical exercise (PE) regimens. Details like the specific fasting intervals, types of exercise, intensity, and duration would add clarity.

Elaborate on the process of randomization and any steps taken to minimize bias in assigning rats to different groups.

2. Clarify the conditions under which the control group was kept. Were they subjected to any standard diet or exercise regimen? This information is crucial for making accurate comparisons.

3. Considering the inclusion of both male and female rats, a gender-specific analysis could be insightful. Investigate whether the effects of IF and PE vary between sexes, as this could have significant implications for the study’s applicability.

4. Compare and contrast your findings with existing literature on IF, PE, and their neurological effects. This will contextualize your results within the broader field of study.

5. Include more detailed graphs or tables to present the behavioral and CSD data. This would aid in the visualization of the results and enhance the reader’s understanding.

6. Ensure that the paper clearly addresses ethical considerations in animal research, including the care and handling of the rats throughout the study.

Comments on the Quality of English Language

Fine

Author Response

Reply to the reviewers of the manuscript BRAINSCI 2744847

(Note: text modifications in the new version of the manuscript are highlighted in yellow color)

Reviewer # 1

- 1. Provide a more comprehensive description of the intermittent fasting (IF) and physical exercise (PE) regimens. Details like the specific fasting intervals, types of exercise, intensity, and duration would add clarity.

Elaborate on the process of randomization and any steps taken to minimize bias in assigning rats to different groups.

Reply: We thank this reviewer by calling our attention to those important methodological points.

Details on fasting methodology are added in lines 90-98, as follows:

“All rats were fed a standard laboratory chow diet (Nuvilab®, with 25% protein) and had free access to water. During six days (PND 25-30), animals were familiarized with the respective feeding/fasting regimes inserted. In the IF group, fasting was started at PND 31 during 24-h for three non-consecutive d/wk [1]. In the IF groups, food was removed from the cage at 08:00 am to start the fasting day and then put back in the cage at the same time the following day to begin the feeding day. The fast was interrupted at the end of eight weeks, and the standard lab-chow diet was thereafter freely available. Body weight and food intake were monitored weekly at the same time in the morning. Food intake was determined from the average consumption found for each cage.”

Details on the exercise protocol are in lines 100-111, as follows:

“Rats were distributed into sedentary and exercised groups; those in the exercise group were familiarized with the treadmill for six days (from PND 25 to 30). Forced physical exercise for rodents (running on a motorized treadmill; INSIGHT, model EP-131) was used. Each familiarization session included placing the rats on the treadmill switched off for 10 minutes and then turning on the treadmill at 8m/min for 5 minutes [13]. The moderate intensity exercise was performed in sessions of 40 minutes a day, three times a week, for eight weeks (from PND31 to 87), according to the following protocol [14]: 5 minutes warming up at a speed of 12 ± 2 m/min, 30 minutes main speed of 20 ± 2 m/min and 5 minutes cooling down at a speed of 12 ± 2 m/min. No aversive stimulus was used, and the rats that refused to run were stimulated with gentle touches; if persistent in refusing, they were discarded from the study. Animals from the sedentary groups were placed on the treadmill switched off for the same period as the trained animals in each session [15].

Details on the randomization are now added in lines 80-82 of the new version of the manuscript, as follows:

“We used a physical method of randomization: pieces of paper with the group’s names written on them were placed in a receptacle, mixed, and withdrawn repeatedly, so each rat was allocated to one of the four groups.”

--------------------------------------------------------

- 2. Clarify the conditions under which the control group was kept. Were they subjected to any standard diet or exercise regimen? This information is crucial for making accurate comparisons.

Reply: We thank the reviewer for having raised this point. As stated in Methods, line 90, "All rats were fed a standard laboratory chow diet (Nuvilab®, with 25% protein) and had free access to water”. The control group was not subjected to IF nor to exercise. (this is stated in line 76-77).

--------------------------------------------------------

  1. Considering the inclusion of both male and female rats, a gender-specific analysis could be insightful. Investigate whether the effects of IF and PE vary between sexes, as this could have significant implications for the study’s applicability.

Reply: We thank this reviewer for raising this important issue. Our analysis revealed sex differences only for body weight and food intake. We included two tables (Table 1 and Table 2) showing those data. This was expected and in line with data from the literature. We have added this information in the new version of the manuscript, in lines 194-195, as follows:

Our findings confirmed data from the literature regarding a higher body weight (Table 1) and greater food ingestion in male rats than females (Table 2).”

On lines 213-215, we further clarified as follows:

“As all groups contained nearly an equal number of males and females, and considering that no difference between sexes was observed in the other parameters (behavior and CSD), we analyzed those parameters by including males and females in a single group.”

--------------------------------------------------------

  1. Compare and contrast your findings with existing literature on IF, PE, and their neurological effects. This will contextualize your results within the broader field of study.

Reply: Thank you for the interesting suggestion. We have compared and contrasted our data with those from the literature. Please refer to the discussion, lines 344-363, as follows:

Another factor that may contribute to the diversity of results is the age of the animals. Unlike most studies in the literature, which use adult or older animals, our study focused on the effects of IF in the development period. This phase is considered critical for neuro-development since the nervous system develops over a long period, from embryonic to puberty, in rats and humans [28]. Environmental factors, including dietary aspects, can interfere with brain development during early life, bringing lasting and often irreversible effects to cognitive development and mental health. Therefore, the IF and PE effects on the developing organism might be considered differently from those effects on the developed brain [29]. While IF and PE appears beneficial in adulthood, the same does not appear to be in developing organisms.

Furthermore, maternal nutrition during pregnancy profoundly affects the growth and development of offspring throughout life [30]. However, the role of IF during pregnancy on fetal development and health is not yet well defined. Even though pregnant Muslim women are exempt from fasting during Ramadan, many of them still choose to participate [31]. For this reason, some animal studies replicate this model of Ramadan fasting during pregnancy. The reported effects are related to an altered profile of placental metabolites, reduced transport of placental amino acids, and fetal growth restriction [32,33]. Impairment of placental function and fetal growth can interfere with development with potentially long-lasting effects on offspring physiology later in life and a greater predisposition to diseases [33].”

--------------------------------------------------------

  1. Include more detailed graphs or tables to present the behavioral and CSD data. This would aid in the visualization of the results and enhance the reader’s understanding.

Reply: The authors thank the reviewer for this relevant suggestion. We have included two tables, replacing one figure (Figure 1). Besides including two tables, we included one figure (now called Figure 1) illustrating the timeline of the various experimental procedures and their respective ages. Furthermore, the remaining four figures (now called Figures 2 to 5) have been modified and now include individual data points, as suggested by reviewer #2.

--------------------------------------------------------

  1. Ensure that the paper clearly addresses ethical considerations in animal research, including the care and handling of the rats throughout the study.

Reply: As stated in lines 65-68, … “This study was approved by the Ethical Committee for using animals in scientific research of the Federal University of Pernambuco (protocol no. 006/2021), whose norms comply with the norms established by the National Institutes of Health Guide for Care and Use of Laboratory Animals (Bethesda, MD, United States).” In addition, in lines 73-74 we included the following statement: All efforts were made to minimize animal suffering and to use the minimum number of animals to get valid results.”

We want to thank the reviewer for the superb reviewing work, which substantially improved our manuscript

Reviewer 2 Report

Comments and Suggestions for Authors

The manuscript by Braz et al examined the impact of intermittent fasting and treadmill exercise on anxiety-like behavior and cortical spreading depression. While this study is of some interest, several points need to be addressed as detailed below:

Major concerns:

1.       Since this study used young rats, the effects of intermittent fasting and treadmill exercise on growth and development should be assessed.

2.       Contrast to published data, the treadmill exercise along had no beneficial effects on anxiety-like behaviors, this is confusing.

3.       The authors mentioned BDNF, what’s the impact of IF and PE on BDNF expression?

Minor concerns:

1.       It is unclear at what time PE was performed.

2.       Is there any sex difference? Are data poured from males and females?

3.       It is unclear when the experiments were performed, the IF and IF+PE groups were under fed or fasted conditions.

4.       The error bars were big for some figures, better to show individual data points.

Author Response

Reply to the reviewer #2 of the manuscript BRAINSCI 2744847

(Note: text modifications in the new version of the manuscript are highlighted in yellow color)

Reviewer # 2

The manuscript by Braz et al examined the impact of intermittent fasting and treadmill exercise on anxiety-like behavior and cortical spreading depression. While this study is of some interest, several points need to be addressed, as detailed below:

Major concerns:

  1. Since this study used young rats, the effects of intermittent fasting and treadmill exercise on growth and development should be assessed.

Reply: The authors thank the reviewer for raising this interesting point. Please refer to Table 1 for these findings. Also, we comment on this point in the discussion, lines 344-353, as follows:

“Another factor that may contribute to the diversity of results is the age of the animals. Unlike most studies in the literature, which use adult or older animals, our study focused on the effects of IF in the development period. This phase is considered critical for neuro-development since the nervous system develops over a long period, from embryonic to puberty, in rats and humans [28]. Environmental factors, including dietary aspects, can interfere with brain development during early life, bringing lasting and often irreversible effects to cognitive development and mental health. Therefore, the IF and PE effects on the developing organism might be considered differently from those effects on the developed brain [29]. While IF appears beneficial in adulthood, the same does not appear to be in developing organisms

--------------------------------------------------------

  1. Contrast to published data, the treadmill exercise along had no beneficial effects on anxiety-like behaviors, this is confusing.

Reply: We thank the reviewer for this interesting observation. In fact, the beneficial effect of our moderate exercise paradigm on behavior was less than one could expect from the literature data. However, there was an effect in the group where exercise was associated with intermittent fasting (group IF+PE); see, for instance, Figure 3-D. The CSD phenomenon also shows a clear effect of exercise (see Figure 5-B). We argue that a more intense or more lasting exercise paradigm probably would produce more conspicuous effects on behavior. (see discussion, lines 374-379).

--------------------------------------------------------

  1. The authors mentioned BDNF, what’s the impact of IF and PE on BDNF expression?

Reply: In absolute agreement with the reviewer, we decide to remove the mention to BDNF, as we did not measure it.

--------------------------------------------------------

Minor concerns:

  1. It is unclear at what time PE was performed.

Reply: We apologize for not clarifying this point in the first version of the manuscript. In the new version, however, we state on lines 100-111:

Rats were distributed into sedentary and exercised groups; those in the exercise group were familiarized with the treadmill for six days (from PND 25 to 30). Physical exercise for rodents (running on a motorized treadmill; INSIGHT, model EP-131) was used. Each familiarization session included placing the rats on the treadmill switched off for 10 minutes and then turning on the treadmill at 8m/min for 5 minutes [13]. The moderate intensity exercise was performed in sessions of 40 minutes a day, three times a week, for eight weeks (from PND31 to 87), according to the following protocol [14]: 5 minutes warming up at a speed of 12 ± 2 m/min, 30 minutes main speed of 20 ± 2 m/min and 5 minutes cooling down at a speed of 12 ± 2 m/min. No aversive stimulus was used, and the rats that refused to run were stimulated with gentle touches; if persistent in refusing, they were discarded from the study. Animals from the sedentary groups were placed on the treadmill switched off for the same period as the trained animals in each session [15].

Please see also the new Figure 1.

--------------------------------------------------------

  1. Is there any sex difference? Are data poured from males and females?

Reply: We thank the reviewer for calling our attention to this point.

As we stated in lines 192-194, Our findings confirmed data from the literature regarding a higher body weight (Table 1) and greater food ingestion in male rats than females (Table 2)”.

In lines 213-215, we stated: “As all groups contained nearly an equal number of males and females, and considering that no difference between sexes was observed in the other parameters (behavior and CSD), we analyzed those parameters by including males and females in a single group.”

--------------------------------------------------------

  1. It is unclear when the experiments were performed, the IF and IF+PE groups were under fed or fasted conditions.

Reply: We thank the reviewer for raising this important point. To help clarify that, we added Figure 1; please refer to this figure.

In addition, details on fasting methodology are added in lines 90-98, as follows:

“All rats were fed a standard laboratory chow diet (Nuvilab®, with 25% protein) and had free access to water. During six days (PND 25-30), animals were familiarized with the respective feeding/fasting regimes inserted. In the IF group, fasting was started at PND 31 during 24-h for three non-consecutive d/wk [1]; food was removed from the cage at 08:00 a.m. to start the fasting day and then put back in the cage at the same time the following day to begin the feeding day. The fast was interrupted at the end of eight weeks, and the standard lab-chow diet was thereafter freely available. Body weight and food intake were monitored weekly at the same time in the morning. Food intake was determined from the average consumption found for each cage.”

Details on the exercise protocol are in lines 100-111, as follows:

“Rats were distributed into sedentary and exercised groups; those in the exercise group were familiarized with the treadmill for six days (from PND 25 to 30). Physical exercise was used for rodents (running on a motorized treadmill; INSIGHT, model EP-131). Each familiarization session included placing the rats on the treadmill switched off for 10 minutes and then turning on the treadmill at 8m/min for 5 minutes [13]. The moderate intensity exercise was performed in sessions of 40 minutes a day, three times a week, for eight weeks (from PND31 to 87), according to the following protocol [14]: 5 minutes warming up at a speed of 12 ± 2 m/min, 30 minutes main speed of 20 ± 2 m/min and 5 minutes cooling down at a speed of 12 ± 2 m/min. No aversive stimulus was used, and the rats that refused to run were stimulated with gentle touches; if persistent in refusing, they were discarded from the study. Animals from the sedentary groups were placed on the treadmill switched off for the same period as the trained animals in each session [15].

--------------------------------------------------------

  1. The error bars were big for some figures, better to show individual data points.

Reply: We thank for this interesting suggestion. Figures 2-5 now contain individual data points in addition to the means and standard variations.

===========================================================

We want to thank the reviewer #2 for the superb reviewing work, which substantially improved our manuscript.

Round 2

Reviewer 1 Report

Comments and Suggestions for Authors

1. Consider adding a brief phrase to introduce the concept of cortical spreading depression (CSD) for readers who may not be familiar with the term.

2. Add specific details about the behavioral indicators of anxiety observed in the IF group, mentioning the affected metrics in the open field (OF) and elevated plus maze (EPM).

3. Clarify "younger rats" by specifying the age range.

4. Briefly discuss the potential significance of the accelerated cortical spreading depression (CSD) in young rats subjected to IF in the context of brain health.

5. Summarize the contrasting effects of IF and PE on anxiety, memory, and CSD in a concise manner.

Comments on the Quality of English Language

Minor editing is required. 

Author Response

(Please note: all modifications are marked in blue color, in the new version of the manuscript.)

Reviewer comment 1- Consider adding a brief phrase to introduce the concept of cortical spreading depression (CSD) for readers who may not be familiar with the term.

Authors reply 1:  We thank the reviewer for this interesting comment. The concept of CSD is now in the lines 51-57, as follows:

CSD consists of a depolarizing “wave-like” reduction (depression) in spontaneous and evoked neuronal activity elicited by an electrical, mechanical, or chemical stimulus at one point of the cerebral cortex. This reversible response spreads slowly concentrically to remote cortical regions while the eliciting point recovers [11]. Considering that the brain's electrical activity controls its primary functions, studying CSD represents an essential tool for understanding brain functioning in health and disease [12].

=======================

Reviewer comment 2 -  Add specific details about the behavioral indicators of anxiety observed in the IF group, mentioning the affected metrics in the open field (OF) and elevated plus maze (EPM).

Authors reply 2:  We thank the reviewer for this comment.  Regarding the OF test, the affected parameters are described in lines 233-237, as follows:

The IF-rats displayed fewer entries into the center (F[3, 60] = 17.328; p < 0.001) and spent a shorter time in the center of the open field (F[3, 60] = 5.646; p = 0.021)). On the other hand, PE increased the number of entries into the center (F[3,60] = 5.646; p = 0,021). There was no statistical difference in time spent in the center between the PE and other groups (Figure 2 C-D).

Regarding the EPM test, the behavioral differences are reported on lines 248-253, as follows (highlighted in blue color in the new version of the manuscript):

ANOVA revealed an interaction between IF and PE regarding the number of entries into the open arms (F[3, 60] = 20.098; p < 0.001) and the time spent in the open arms (F[3,60] = 9.101; p = 0.004). Intermittent fasting reduced the number of entries and time spent in the open arms compared to the C group and IF+PE group (p < 0.05). PE was associated with increased entries into the open arms and long time spent in the open arms, which was more intense in the group IF+PE.

====================================

Reviewer comment 3 - Clarify "younger rats" by specifying the age range.

Authors reply 3:  We thank the reviewer for calling our attention to this point. The age range is now specified in the abstract (line 15), and in the legend of Figure 1 (lines 188-190), as follows:

"We classified the animals as young because the rats were subjected to IF and PE at 31-to-87 PND."

=============================================

Reviewer comment 4 - Briefly discuss the potential significance of the accelerated cortical spreading depression (CSD) in young rats subjected to IF in the context of brain health.

Authors reply 4:  The authors thank very much the reviewer for raising this important point. In lines 373-376 we have briefly discussed it, as follows:

"Nutritional deficiency early in life facilitates CSD propagation in the rat cortex (12). In the present study, the lower body weight and food intake observed in the IF group (Tables 1 and 2) suggest a certain degree of nutritional deficiency, which is in line with the CSD facilitation in these animals (Figure 5)."

===================================

Reviewer comment 5 -  Summarize the contrasting effects of IF and PE on anxiety, memory, and CSD in a concise manner.

Authors reply 5: - In agreement with the reviewer, we summarized the contrasting effects of IF and PE, in lines 394-396, as follows:

"Summarily, IF in young rats worsens anxiety-like behavior and memory and accelerates CSD, while PE improves the behavioral aspects of anxiety and memory and slows CSD."

The authors thank the nice reviewing work done by the reviewer, which improved very much the quality of our manuscript.

Reviewer 2 Report

Comments and Suggestions for Authors

The authors have done a good job in addressing all my concerns.

Author Response

Reviewer's Comment: "The authors have done a good job in addressing all my concerns."

AUTHORS'  REPLY: The authors thank the reviewers for their helpful and generous contribution to improving our manuscript.